# An Exploration of a Reflective Evaluation Tool for the Teaching Competency of Pre-Service Physical Education Teachers in Korea

**Chul-Min Kim and Eun-Chang Kwak \***

Department of Physical Education, Kyung Hee University, Yongin-si 17104, Gyeonggi-do, Korea; cmkim@khu.ac.kr
\* Correspondence: eckwak@khu.ac.kr; Tel.: +82-10-2459-3578

**Abstract:** The purpose of this study was to develop an evaluation tool that could help pre-service teachers develop their teaching skills. As for the research method, the Delphi technique was used to collect opinions from physical education professor evaluation experts. The survey was conducted three times and various opinions of experts were collected and analyzed. The newly constructed evaluation tool consists of 46 questions for class preparation (the creation of the learning environment), the introduction (routine activities, learning goals, and task presentation), development (class strategy, observation and interaction, and the maintenance of the learning environment), and conclusion (routine activities, summary, and closure). It was designed to increase the accuracy of evaluation by developing evaluation criteria for each question. An evaluation tool including quantitative and qualitative methods for use in pre-service physical education teacher education was developed. The significance of this study is the development of an effective evaluation tool that can evaluate the core teaching behaviors in the field of physical education. This evaluation tool should be used as a learning tool that includes planning, operation, evaluation, and seeking improvement measures through reflective activities. If pre-service teacher education institutions apply this evaluation tool in their teacher training programs, it would be a great chance to learn how to develop and sustain teaching abilities and effectiveness.

**Keywords:** reflective evaluation tool; pre-service teacher education; teaching competency; pre-service physical education teacher; Delphi survey



## 1. Introduction

Teaching competency greatly affects students' academic achievement, the creation and maintenance of an efficient learning environment, the provision of appropriate learning tasks for learners' needs, and active class management [1]. Physical education classes require more systematic and professional teaching skills for various class environments, such as playgrounds and gymnasiums. However, many physical education teachers lack basic teaching skills, such as creating a teaching environment, presenting tasks, and utilizing various teaching tools [2,3]. Beginner teachers who have just graduated from pre-service teacher education programs tend not to recognize the various situational variables of teachers, students, and the curriculum [4]. Pre-service teachers have a sense of the gap between the knowledge learned in pre-service teacher education programs and the real world, and experience confusion and various shocks from real-world teaching [5]. This may come from the failure of pre-service teacher education programs in teaching skills that can be directly used in the field. The cultivation of systematic teaching competencies should be the core of pre-service teacher education programs [6]. For example, Grossman and McDonald [7] emphasized that pre-service teachers should be able to develop class analysis, critical thinking, and self-reflection skills through practical experiences. Graham [8] also proposed that teaching skills could be cultivated through thinking reflectively about class or

teaching performance. As such, the teaching competency of prospective physical education teachers has many problems regarding class management, class planning, task presentation, feedback, the creation of a learning environment, and the process of reflective activities.

In fact, pre-service teacher education programs should reduce the gap between theory and practice and provide practice-oriented education based on reflections on teaching [9,10]. The most effective way to improve teaching competency through reflective teaching activities is by reflecting on schools in the real world [11]. However, it is not easy to objectively evaluate teaching competency because the concepts and standards for effective teaching methods are ambiguous [12,13]. The evaluation of a teacher's teaching competency and reflective teaching activities for academic achievement also bring about important changes in the teacher's qualities and thinking [14]. In particular, the importance of evaluation tools that can be objectively and systematically evaluated for flexible class management increases. Therefore, pre-service teacher education programs should be operated so that pre-service teachers can grow through practice and reflection, along with the evaluation of their teaching competency.

Surprisingly, it is not easy to develop a tool to evaluate the teaching competency of physical education teachers. This is because in order to evaluate teaching competency, standards for teaching performance must be clearly prepared, and the scope and sequence must be appropriate. It is important to secure validity and reliability in an evaluation tool for actual physical education classes. In particular, it has been found that pre-service teacher education institutions in Korea do not provide sufficient educational opportunities or experiences of teacher evaluation to pre-service teachers [15]. Recently, Kim [15] developed a tool for evaluating physical education teachers' performance and evaluated middle school physical education teachers in Korea. He found that teachers with 11–20 years of teaching experience achieved the highest score, teachers with 0–5 years of teaching experience achieved the second highest score, and teachers with more than 21 years of teaching experience achieved the lowest score. This study is meaningful in that it accurately identified problems related to evaluation in the field by evaluating physical education teachers based on objective criteria.

It has been consistently argued that practice-oriented education is necessary to systematically equip teachers with teaching skills through pre-service teacher education programs [15–17]. However, the lack of basic teaching competency in physical education teachers shows the justification to make all-out efforts to improve the teaching competency in pre-service teacher education programs. Efforts to improve teaching competency in pre-service teacher education programs are not unique to Korea. In particular, the purpose of teacher education is to cultivate excellent quality teachers in any country, regardless of region, economy, or environment. In order to guarantee the validity and reliability of the teacher education program, the quality of pre-service teachers is very important. The quality of teachers cannot be guaranteed by simply graduating from university, or even after entering the field; they must be able to continue to perform their jobs and grow into sustainable teachers [18]. In other words, teachers should continuously secure the competitiveness of their teaching skills and show their growth as experts through thorough evaluation. In this process, the understanding and development of evaluation tools in pre-service teacher education offers insight into the course of class management and fosters teaching competency, along with the reflection process. Nevertheless, there are very few studies that have developed, applied, and confirmed teaching behavior-related evaluation tools. This may be because teachers are reluctant to implement the evaluation itself because they are concerned about the results of the evaluation. However, if it is self-evaluation rather than evaluation by others, it is not too burdensome or worrisome to reflect on one's own class and seek improvements. Therefore, the purpose of this study was to develop an evaluation tool that can more likely be used in a gym context by developing evaluation questions and standards related to the teaching ability of pre-service physical education teachers.

## 2. Theoretical Framework

### 2.1. Teaching Functions in Physical Education

Physical education subjects require more diverse teaching skills than general teachers due to the specificity of education that should be taught in consideration of learners' physical characteristics [19]. Rink [2] also presented elements of 'clear task presentation', 'group, time, learning space, tool organization', 'creation and maintenance of learning environment', 'teacher behavior', 'class plan', and 'evaluation' as teaching functions required by physical education teachers. Manross and Templeton [20] said that the level of professionalism of physical education teachers can be grasped through class planning, individual student guidance strategies, perceptual ability, automated teaching behavior, impromptu and creative feedback, appropriate use of subject knowledge, and reflective inquiry ability.

Physical education teachers are very important for cultivating the ability to learn and practice basic teaching functions. Therefore, it is necessary to provide practice-oriented education that can have basic teaching functions from pre-service teacher education. However, pre-service teacher education is limited in providing sufficient opportunities to practice teaching functions. It does not guarantee the quality of education for fostering competent teachers that is necessary in the school field. Therefore, the pre-service teacher education program is needed to systematically improve teaching functions. This will help to cultivate pre-service teachers' teaching ability so that they can minimize the gap between theory and provide beneficial physical education classes to students.

### 2.2. Characteristics and Limitations of Physical Education Teacher Evaluation Research

Evaluation studies have been conducted for a long time to cultivate the training of physical education teachers' [12,13,15,21–25]. The study of teaching behavior analysis was introduced in the 1970s, which helped improve the teaching efficiency and physical education teaching competency of teachers [26]. Research to evaluate teachers' behavior through systematic observation and analysis methods provided a lot of help in improving teaching ability. However, there was a problem that only certain teaching functions, such as teacher behavior, language, and feedback, could be evaluated, and it took a lot of time to grasp the evaluation process and results [27,28]. Rink and Werner [22] developed QMTPS to analyze the type of task, task presentation, student response to the task, and teacher behavior through feedback to improve the quality of teaching competency. However, research related to the evaluation of teaching competency has the problem of being inefficient for practical use in the field because it focuses only on selected factors, such as teacher behavior, language, feedback, or evaluation [23]. For example, NASPE [24] developed a physical education teacher evaluation tool to pursue psychomotor, cognitive, and affective values, and to use various teaching methods and strategies. Rink [23] emphasized the validity and efficiency of the evaluation process and results by comparing the evaluation tools used in physical education and general academic fields. Meanwhile, the GDE (Georgia Department of Education) [25] used an evaluation method that employs rubrics based on planning, learning evaluation, learning environment, and the professional domain.

In order to develop evaluation tools that are highly utilized in schools, it is necessary to refer to the qualification criteria for physical education teachers, such as NASPE (National Association for Sport and Physical Education), SHAPE America (Society of Health and Physical Education), and NBPTS (National Board for Professional Standards). First, NASPE [29] develops professional standards for physical education teachers and presents standards for efficient class management. Specifically, it deals with the effects and expertise of science and theoretical knowledge, exercise function and physical strength, class planning and operation, class and management, and student achievement. SHAPE America [30] presents the qualification criteria for physical education teachers with subject content and basic knowledge, skills and health-related physical strength, planning and performance, class management and learning motivation, academic achievement and evaluation, and professional responsibility. NBPTS [31] also consists of content similar to the qualification

standards of NASPE and SHAPE America, but the criteria are presented in more detail. Each institution presents evaluation indicators for the evaluation of qualification standards and the expertise of physical education teachers. However, although the overall content of physical education classes is comprehensively dealt with, it is difficult to use these criteria as tools to evaluate practical teaching skills due to a lack of specificity. For example, RATE (Rapid Assessment of Teacher Effectiveness) has its own advantages in distinguishing efficient teacher behavior by evaluating the guidance of learning goals, effective class pacing, and feedback [21]. However, it does not fit the characteristics of the subject of physical education and it is difficult to evaluate factors, such as teacher–student interaction and class management when limiting the teaching behavior evaluation criteria to 10.

Previous studies related to teacher evaluation have made continuous efforts to find tools optimized for the evaluation of physical education teachers. However, most physical education teacher evaluation tools are used to evaluate general class teachers, and physical education teaching competency is not professionally evaluated, which remains a limitation. There are considerable difficulties for beginners or pre-service teachers who tend to have a lack of experience and knowledge in evaluation methods. For this reason, teachers may be passive or reluctant to use evaluation tools. In addition, there is difficulty in assessing the psychomotor, cognitive, and affective domains, which are important learning goals of school curricula [12]. For this reason, research on the evaluation of physical education teachers has continued.

It is necessary to pay a lot of attention to pre-service teacher education research. Pre-service teacher education is a very important role to form the correct sound values and beliefs as a teacher. Therefore, evaluation tools for pre-service teacher education should be developed and utilized in various teaching situations in the pre-service teacher education programs. It should include the process of class operation, evaluation, and reflection to form a perspective of class and systematic teaching ability.

### 3. Method

#### 3.1. Delphi Method

There are very limited opportunities to teach in pre-service teacher education programs. Therefore, specific efforts are needed to ensure a certain level of teaching skills. Pre-service teachers often do not have specific teaching skills, such as how to create a learning environment or how to teach. In this case, the Delphi technique for collecting various opinions from experts and preparing evaluation criteria can effectively be used. The Delphi survey presents effective measures by integrating the experience, knowledge, and know-how of related experts to solve a given problem [32,33].

The Delphi technique secures homogeneity in the heterogeneous opinions or judgments of experts and finally proves its validity by deriving results based on consensus [34,35]. Therefore, it is the most effective investigation method of systematically collecting and organizing expert opinions in developing evaluation tools [36]. The Delphi method was used to collect various opinions from sports education experts to develop a reflective teaching competency evaluation tool for pre-service teachers.

#### 3.2. Delphi Method Participants

Delphi surveys are conducted with a variety of universality, valid criteria, and sample sizes. Therefore, in this study, we tried to balance the response rate with experts and secure validity. Four university professors and six physical education teachers were recruited. Specifically, the professors were Ph.D. holders in sports pedagogy and had at least five years of teaching experience as physical education teachers or were experts in teaching methods and teaching evaluation. The physical education teachers were limited to teachers with more than seven years of teaching experience as physical education teachers in middle and high schools and were interested in developing their teaching expertise in areas, such as teaching and learning programs and educational technology.

### 3.3. Delphi Method Procedure

The Delphi survey was conducted three times using expert panels. The first time, we conducted an open survey to review evaluation questions and collect opinions for evaluation criteria. The evaluation criteria were limited to the teaching behavior and learning environment of teachers directly shown in physical education classes. It was advised not to present opinions on factors that could not be directly evaluated in class. In addition, opinions were guided to be suggested based on the national physical education curriculum in Korea. The survey results were reviewed with three experts (one professor from the Department of Physical Education and two doctors of sports education). They reviewed the draft survey, found problems, shared opinions, and created the first Delphi survey tool. Then, after classifying the expert opinions obtained through the first survey, a draft of the evaluation tool was prepared, and the second Delphi survey was conducted. The second and third Delphi surveys reviewed the appropriateness of the questions and evaluation criteria.

### 3.4. Questionnaire Development

In this study, the physical education teaching competency evaluation tool developed by Kim [15] was used. This evaluation tool was developed based on various evaluation tools, such as the Physical Education Teacher Evaluation Tool [24] and the Framework for Teaching Evaluation Instrument [37]. This tool specifically presents the core teaching behavior required based on the stage of the class. Therefore, systematic and effective evaluation makes it possible to develop a learning tool to cultivate the teaching competency of pre-service teachers. Evaluation criteria for each question on teaching behavior were developed and modified so that knowledge of the teaching method could be effectively learned. The first Delphi survey collected opinions on the content system of core teaching functions, evaluation criteria for each question, and the development of a reflective journal. The second Delphi survey produced a draft reflective teaching competency evaluation tool by analyzing the expert opinions collected in the first round. Additionally, an expert evaluation was conducted to confirm the suitability of each question. A five-point Likert scale was used to confirm the suitability of each question. Additional opinions for the production of a reflective journal were also collected. Finally, the third Delphi survey reviewed the results of the second Delphi survey, and finally confirmed the suitability of the questions and evaluation criteria. Overall, in order to collect expert opinions as widely as possible, if opinions between experts were in disagreement from the beginning of the study, homogeneity was secured, and the implications were finally reached.

### 3.5. Data Analysis

In the first Delphi survey, collected expert opinions were analyzed by the inductive category analysis method. In the second and third Delphi surveys, the suitability of the questions was confirmed through the results of the mean, standard deviation, content validity, and the coefficient of variation. The content validity ratio (CVR) was composed of questions with a measured value of 0.62 or more according to the appropriate verification criteria, i.e., the frequency with which the evaluation criteria and indicators were responded to as valid [38]. Questions with a CV score of less than 0.50 were reviewed for adequacy, and if there were any problems with the contents, they were deleted [39].

### 3.6. Ethics

In this study, it was very important to present the process of developing evaluation tools by conducting a survey on expert panels. First, the purpose of the study and the role of the experts were explained by phone or face-to-face based on the expert's situation, and IRB consent was obtained. All experts were informed that if it was difficult to read the contents of the research ethics agreement and participate in the study, participation can be stopped at any time, and it was indicated that there was no disadvantage in doing so. The Delphi survey data were shared through the Kakao Talk message program (social network

system in Korea). All of the collected data were only used for this study, and personal information was not used. Then, the data were analyzed and the derived contents were reviewed (member checks) between the expert panel and the members. In addition, peer debriefing and triangulation were conducted for data analysis with two Ph.D. students in sports pedagogy who had experience in teacher evaluation and teaching methods research. We tried to reveal only the factual aspects of the results to prevent distortion of the research results. This study was conducted after obtaining IRB approval (GINUEIRB-2021-005) from Gyeongin National University of Education in Korea.

## 4. Results

### 4.1. First Round

In the first Delphi survey, expert opinions were collected for the development of reflective teaching competency evaluation tools for pre-service physical education teachers. It was confirmed that the actual training of teaching competency is the most important evaluation tool for the pre-service physical education teacher education program. Based on the existing teaching performance evaluation tool, a reflective teaching competency evaluation tool for pre-service physical education teachers was developed involving four stages: class preparation, introduction, development, and conclusion. During the review of the evaluation questions, the relevant questions were included or deleted after confirming that the contents of the two questions were duplicated or unnecessary. The tool consists of "class preparation," i.e., the creation of the learning environment (five questions); "introduction," i.e., routine activities (five questions); learning goals and task presentation (10 questions); "development," i.e., class strategy (seven questions); observation and interaction (six questions); the maintenance of the learning environment (seven questions); "conclusion," i.e., routine activities (three questions); and summary and closure (five questions). In total, this evaluation tool was composed of 48 questions involving four stages and eight domains.

Subsequently, according to the expert opinions, the evaluation criteria for each question and the framework for writing a reflection journal were constructed. The reflection journal was created to write down what was lacking, what was good, what needed improvement, and what was felt throughout the evaluation based on the evaluation results. This evaluation tool applies both quantitative and qualitative evaluation methods so that pre-service teachers can understand the meaning of each evaluation question and reflect on the objective evaluation and results. The results of the first Delphi survey were reflected in the production of the second Delphi survey tool. The results of the first round of the Delphi survey are presented in Tables 1 and 2.

**Table 1.** Results of the first Delphi survey.

| Stage | Domains | Questions | | |
|---|---|---|---|---|
| | | **Agree** | | **Disagree** |
| 1. Preparing for class | Creating a learning environment | 1-1-1. Data on a suitable place<br>1-1-2. Securing teaching material and spaces<br>1-1-3. Safety inspection of the learning place | 1-1-4. Preparing for the learning materials<br>1-1-5. Creating an enjoyable class atmosphere | |
| 2. Introduction | 2-1. Routine activity (start) | 2-1-1. Attendance and uniform check<br>2-1-2. Health check<br>2-1-3. Warm-up | 2-1-4. Using the rules<br>2-1-5. Smooth progress | |
| | 2-2. Learning goals and task presentation | 2-2-1. Attention<br>2-2-2. Recall of previous learning contents<br>2-2-3. Appropriateness of learning goals and tasks<br>2-2-4. The clarity of task presentation<br>2-2-5. Use of demonstrations, media, and cues | 2-2-6. Use of appropriate language<br>2-2-7. Motivation<br>2-2-8. Use of various questions<br>2-2-9. Giving students a role<br>2-2-10. Safety education | |
| 3. Development | 3-1. Class strategy | 3-1-1. Use of various teaching and learning methods<br>3-1-2. Providing tasks that consider the characteristics of learners<br>3-1-3. Integrated operation of learning content<br>3-1-4. Teaching method that considers learner characteristics | 3-1-5. Promoting understanding through demonstrations, media, and cues<br>3-1-6. Providing tasks based on the level of development of the tasks<br>3-1-8. Checking the progress of learning | 3-1-7. Organizing and guiding learning |
| | 3-2. Observation and interaction | 3-2-1. Providing feedback<br>3-2-2. Using a questionnaire<br>3-2-3. Creating an atmosphere for communication<br>3-2-4. Inducing interaction with others | 3-2-5. Verbal and nonverbal communication<br>3-2-6. Fair and equal treatment | |
| | 3-3. Maintaining the learning environment | 3-3-1. Appropriateness of the place based on the activity<br>3-3-2. Appropriateness of the learning organization<br>3-3-3. Efficient control and operation of class hours<br>3-3-4. Providing sufficient learning time | 3-3-5. Sufficient use of teaching material or media<br>3-3-6. Inappropriate behavior during instruction<br>3-3-7. Securing the continuous safety of the learning environment | |
| 4. Conclusion | 4-1. Routine activity (finish) | 4-1-1. Cool down<br>4-1-3. Patient check<br>4-1-4. Smooth progress | | 4-1-2. Organizing learning materials |
| | 4-2. Summary and closure | 4-2-1. Confirmation of the understanding of learning contents<br>4-2-2. Learning process and outcome evaluation | 4-2-3. Encouraging students<br>4-2-4. Learning transfer<br>4-2-5. Previewing the next lesson | |

**Table 2.** Deleted questions according to the results of the first Delphi survey.

| Stage | Domains | Questions | Expert Opinion | Note |
|---|---|---|---|---|
| 3. Development | 3-1. Class strategy | 3-1-7. Is the organization of the learning (individual or group) appropriate? | -Rules can be more effective when applied in advance <br> -It should be included in the questions related to the smooth development of routine activities | Delete |
| 4. Conclusion | 4-1. Routine activity (finish) | 4-1-2. Do you give and guide students (groups) roles to organize the learning materials? | It should be included as a question related to class rules or routine activities | Delete |

*4.2. Second and Third Rounds*

The second Delphi survey was in the stage of reviewing the draft evaluation tool produced through the first Delphi survey. The draft evaluation tool identified descriptive statistics, content validity (CVR), and the validity index (CV) to evaluate the appropriateness of the evaluation questions and evaluation criteria for each question. The content validity index of the question showed a level from 0.8 to 1.0, confirming the appropriateness of the question. However, two questions were deleted because they did not meet the criteria for the data analysis: "Do you determine and use various rules necessary for classes?" in the domain of learning goals and task presentation (M:4.30, SD:1.06, CVR:0.6, CV:0.25), and "Is the teaching method that reflects the characteristics of the school or learner properly used?" in the teaching strategy domain (M:4.00, SD:1.05, CVR:0.4, CV:0.26). In the second Delphi survey, the evaluation tool consisted of a total of 46 questions. The third Delphi survey was conducted in the same way as the second survey. The appropriateness of the revision of the questions and evaluation criteria was evaluated based on the results of the second Delphi survey. All the questions showed content validity index scores from 0.8 to 1.0, and a CV index from 0.06 to 0.15, confirming the appropriateness of the evaluation tool. The reflective journal agreed with all of the secondary expert opinions and there were no amendments. The questions, evaluation criteria, and reflection journal form were reviewed to increase the completeness of the evaluation tool. The results of the second and third rounds of the Delphi survey are presented in Tables 3 and 4.

**Table 3.** Deleted questions based on the second results.

| Stage | Domains | Questions | Expert Opinion | Note |
|---|---|---|---|---|
| 2. Introduction | 2-1. Routine activity (start) | 2-1-4. Do you set and use the rules necessary for the class? <br><br> There are rules and student roles for various activities such as warm-up exercises, learning material preparation, and group formation <br><br> There are no set rules and the teacher presents extempore | It should be presented as an example of student roles and routine activities or evaluation criteria | Included in 2-2-10 |
| 3. Development | 3-1. Class strategy | 3-1-2. Are teaching and learning methods that reflect the characteristics of schools or learners appropriately utilized? <br><br> A teaching method reflecting the region and culture of the school is applied <br><br> General class with no specific meaning | -The difference from other questions related to student characteristics is unclear <br> -The evaluation criteria are ambiguous-More specific content is needed | Included in 3-1-4 |

**Table 4.** The CVR results for each question according to the second and third Delphi surveys.

| | | | Round 2 | | | | Round 3 | | | |
|---|---|---|---|---|---|---|---|---|---|---|
| Stage | Domains | Questions | M | SD | CVR | CV | M | SD | CVR | CV |
| 1 | 1-1 | 1 | 4.6 | 0.52 | 1.00 | 0.11 | 4.8 | 0.42 | 1.00 | 0.09 |
| | | 2 | 4.6 | 0.52 | 1.00 | 0.11 | 4.9 | 0.32 | 1.00 | 0.06 |
| | | 3 | 4.7 | 0.48 | 1.00 | 0.10 | 4.7 | 0.67 | 0.80 | 0.14 |
| | | 4 | 4.6 | 0.52 | 1.00 | 0.11 | 4.4 | 0.52 | 1.00 | 0.12 |
| | | 5 | 4.3 | 0.67 | 0.80 | 0.16 | 4.5 | 0.53 | 1.00 | 0.12 |
| 2 | 2-1 | 1 | 4.2 | 0.63 | 0.80 | 0.22 | 4.4 | 0.52 | 1.00 | 0.12 |
| | | 2 | 4.3 | 0.95 | 0.80 | 0.22 | 4.7 | 0.48 | 1.00 | 0.10 |
| | | 3 | 4.2 | 0.92 | 0.80 | 0.22 | 4.9 | 0.32 | 1.00 | 0.06 |
| | | 4 | 4.3 | 1.06 | 0.60 | 0.25 | | | | |
| | | 5 | 4.6 | 0.52 | 1.00 | 0.11 | 4.7 | 0.48 | 1.00 | 0.10 |
| | 2-2 | 1 | 4.7 | 0.48 | 1.00 | 0.10 | 4.8 | 0.42 | 1.00 | 0.09 |
| | | 2 | 4.1 | 0.88 | 0.80 | 0.21 | 4.8 | 0.42 | 1.00 | 0.09 |
| | | 3 | 4.2 | 0.92 | 0.80 | 0.22 | 4.9 | 0.32 | 1.00 | 0.06 |
| | | 4 | 4.6 | 0.70 | 0.80 | 0.24 | 4.8 | 0.42 | 1.00 | 0.09 |
| | | 5 | 4.5 | 0.97 | 0.80 | 0.37 | 4.8 | 0.42 | 1.00 | 0.09 |
| | | 6 | 4.6 | 0.52 | 1.00 | 0.11 | 4.8 | 0.42 | 1.00 | 0.09 |
| | | 7 | 4.4 | 0.70 | 0.80 | 0.16 | 4.9 | 0.32 | 1.00 | 0.06 |
| | | 8 | 4.5 | 0.71 | 0.80 | 0.16 | 4.8 | 0.42 | 1.00 | 0.09 |
| | | 9 | 4.4 | 0.70 | 0.80 | 0.35 | 4.6 | 0.70 | 0.80 | 0.15 |
| | | 10 | 4.4 | 0.70 | 0.80 | 0.16 | 4.7 | 0.48 | 1.00 | 0.10 |
| 3 | 3-1 | 1 | 4.8 | 0.42 | 1.00 | 0.09 | 4.8 | 0.63 | 0.80 | 0.13 |
| | | 2 | 4.0 | 1.05 | 0.40 | 0.26 | | | | |
| | | 3 | 4.8 | 0.42 | 1.00 | 0.09 | 4.8 | 0.42 | 1.00 | 0.09 |
| | | 4 | 4.4 | 0.70 | 0.80 | 0.32 | 4.6 | 0.52 | 1.00 | 0.11 |
| | | 5 | 4.4 | 0.70 | 0.80 | 0.40 | 4.8 | 0.42 | 1.00 | 0.09 |
| | | 6 | 4.4 | 0.70 | 0.80 | 0.16 | 4.8 | 0.42 | 1.00 | 0.09 |
| | | 7 | 4.5 | 0.97 | 0.80 | 0.22 | 4.7 | 0.48 | 1.00 | 0.10 |
| | 3-2 | 1 | 4.3 | 0.95 | 0.80 | 0.22 | 4.7 | 0.48 | 1.00 | 0.10 |
| | | 2 | 4.6 | 0.52 | 1.00 | 0.11 | 4.9 | 0.32 | 1.00 | 0.06 |
| | | 3 | 4.2 | 0.92 | 0.80 | 0.22 | 4.8 | 0.42 | 1.00 | 0.09 |
| | | 4 | 4.5 | 0.71 | 0.80 | 0.16 | 4.6 | 0.52 | 1.00 | 0.11 |
| | | 5 | 4.8 | 0.42 | 1.00 | 0.09 | 4.7 | 0.48 | 1.00 | 0.10 |
| | | 6 | 4.7 | 0.48 | 1.00 | 0.10 | 5.0 | 0.00 | 1.00 | 0.00 |
| | 3-3 | 1 | 4.4 | 0.70 | 0.80 | 0.16 | 4.6 | 0.52 | 1.00 | 0.11 |
| | | 2 | 4.3 | 0.67 | 0.80 | 0.25 | 4.6 | 0.52 | 1.00 | 0.11 |
| | | 3 | 4.6 | 0.52 | 1.00 | 0.11 | 4.7 | 0.48 | 1.00 | 0.10 |
| | | 4 | 4.8 | 0.42 | 1.00 | 0.19 | 4.9 | 0.32 | 1.00 | 0.06 |
| | | 5 | 4.4 | 0.70 | 0.80 | 0.16 | 4.7 | 0.48 | 1.00 | 0.10 |
| | | 6 | 4.7 | 0.48 | 1.00 | 0.10 | 4.9 | 0.32 | 1.00 | 0.06 |
| | | 7 | 4.4 | 0.70 | 0.80 | 0.16 | 4.8 | 0.42 | 1.00 | 0.09 |
| 4 | 4-1 | 1 | 4.3 | 0.67 | 0.80 | 0.16 | 4.8 | 0.42 | 1.00 | 0.09 |
| | | 2 | 4.6 | 0.70 | 0.80 | 0.19 | 4.7 | 0.48 | 1.00 | 0.10 |
| | | 3 | 4.6 | 0.52 | 1.00 | 0.11 | 4.4 | 0.52 | 1.00 | 0.12 |
| | 4-2 | 1 | 4.5 | 0.71 | 0.80 | 0.19 | 4.7 | 0.48 | 1.00 | 0.10 |
| | | 2 | 4.3 | 0.67 | 0.80 | 0.16 | 4.6 | 0.52 | 1.00 | 0.11 |
| | | 3 | 4.6 | 0.52 | 1.00 | 0.11 | 4.5 | 0.53 | 1.00 | 0.12 |
| | | 4 | 4.3 | 0.67 | 0.80 | 0.16 | 4.8 | 0.42 | 1.00 | 0.09 |
| | | 5 | 4.6 | 0.52 | 1.00 | 0.11 | 4.8 | 0.42 | 1.00 | 0.09 |

*4.3. Final Reflective Evaluation Tool's Questions and Criteria*

An evaluation tool was developed by conducting three round of Delphi surveys to evaluate the appropriateness of the questions and analyze the opinions of experts. The composition of the evaluation tool for each domain is as follows. First, creating a learning environment (five questions) in the class preparation stage emphasizes the competency to select a suitable place for pre-class learning content, secure teaching tools and facilities and arrange them appropriately, create a safe and enjoyable learning environment, and

provide positive class participation opportunities. Second, the introduction stage consists of the domains of starting a routine activity (2-1), learning goals, and task presentation (2-2). Routine activities (four questions) evaluate students' competency to efficiently perform repetitive behaviors at the beginning of classes, such as checking student attendance and health. Learning objectives and task presentation (10 questions) emphasize the competency to create an efficient understanding of learning contents and to induce active class participation by creating a learning environment suitable for learning objectives and task delivery. Third, the development stage consists of the domains of class strategy (3-1), observation and interaction (3-2), and maintaining the learning environment (3-3). The class strategy (five questions) evaluates the competency to provide meaningful learning value to learners by planning learning goals and learning contents that meet the characteristics and needs of students and utilizing appropriate teaching strategies. Observation and interaction (seven questions) represent the competency to communicate by providing appropriate explanations, feedback, and various questions so that all students can positively participate in class. Maintaining the learning environment (seven questions) emphasizes the evaluation of the competency to operate classes smoothly by appropriately adjusting student organization and management, time, space, teaching materials, and facilities. The fourth stage, the conclusion, consisted of a routine activity (4-1) and a summary and closure (4-2). Routine activities (three questions) emphasize the evaluation of the ability to efficiently operate repetitive behaviors in the class conclusion stage, such as organizing materials and checking health status. Summary and closure (five questions) emphasize the evaluation of students' competency to provide appropriate information by evaluating their understanding of learning goals, achievement, and the process of participating in tasks. Lastly, the reflection log is structured so that students can reflect on their lessons in depth by analyzing the evaluation results for each stage and writing the good points, the bad points, the points to be improved, and the points felt through the evaluation tool.

## 5. Discussion

A reflective evaluation tool was developed to improve the teaching competency of pre-service physical education teachers in this study. The evaluation tool enables detailed inspection through all class courses from preparation to completion. Therefore, it is expected to contribute to forming a perspective on classes and improving teaching skills. The specific characteristics of the evaluation tool are discussed as follows.

First, the "class preparation" stage evaluates the competency to prepare for the planning process for class management before class starts. Creating a learning space and preparing learning materials based on the class plan helps smooth class progress [40]. It is possible to promote an understanding of the class by providing a range of learning information to students even before the class starts. Therefore, in the domain of class preparation in reflective evaluation tools, the ability of learners to create a learning environment for active participation is required. The evaluation questions consist of the selection of appropriate learning places, teaching materials and spaces, safety checks, the preparation of learning materials, and the creation of a positive class atmosphere. The content validity was from 0.8 to 1.0, showing the appropriateness of the questions. This stage also emphasizes safety checks, the preparation of learning materials, and the creation of a positive class atmosphere. This shows that content areas that are generally easy for pre-service teachers to neglect are being dealt with. Pre-service teachers are well aware of preparing teaching materials to create a learning environment. However, they have low awareness of safety and creating a pleasant atmosphere [41]. They do not understand the nature of a learning environment. The more thoroughly the teacher prepares for the class, the higher their level of interaction with the students and quality of the class [42]. Therefore, the class preparation stage is characterized by presenting conditions for the competency to create a learning environment that can increase students' participation, along with various subject knowledge.

Second, the "introduction stage" is a domain that evaluates the competency of teachers to effectively deliver organizational learning content. This stage is divided into routine activities, learning goals, and task presentation domains. Routine activities consist of questions about attendance and uniform checks, health status checks, warm-up exercises, the use of rules, and smooth progress. The content validity of all questions appeared to be 1.0, which is considered very appropriate. Routine activities are class strategies to effectively perform repetitive activities conducted in each class, such as warm-up and attendance checks [2]. Kim [15] reported that patient identification and warm-up exercises were found to be at a normal level of 3.10 out of 5 points, which means that teachers are not properly conducting routine activities. Routine activities can maximize the time for learning activities by minimizing class management time. Therefore, the class management strategy to provide maximum learning time for students by reducing management time in each class is very important. Thus, there is a need to focus on developing rules and procedures for efficient class management and tools to evaluate a teacher's ability to systematically conduct a lesson [43]. Learning goals and task presentation consist of student attention, the recall learning of previous class content, consistency between learning goals and tasks, the clarity of task presentation, motivation, various questions, student roles, and safety education. The content validity was found to be at the very appropriate level of 1.0, except for 1 (0.8) out of 10 questions. In this domain, the rate of questions was the highest in the evaluation tool. It can be said that understanding the learning content is very important for increasing the value of continuous participation and learning in class. A good class is a class that sufficiently achieves the learning goals expected of students and leads students to actively participate in making themselves feel satisfied and enjoy the class. Learning goals and learning tasks should be presented together, not separately. Kim [15] reported that physical education teachers evaluated learning goals and presentation of assignment scores were 2.82 out of 5 points, which were found to be very insufficient. If learning goals and tasks are not accurately presented, not only will the direction of the class be lost, but it can also negatively affect students' academic achievement due to them not understanding the learning content. Therefore, this domain provides a basis for pre-service teachers to recognize the need for and practice with presenting learning goals and tasks in class.

Third, the "development stage" evaluates the competency of teachers to continue successful classes by checking and inducing students to participate in the activity based on the class plan. In this evaluation tool, the development stage consists of sub-domains of instructional strategy, observation and interaction, and the maintenance of the learning environment. The class strategy consists of the appropriateness of teaching and learning methods, integrated composition and guidance of learning contents, covered guidance with regard to learner characteristics, guidance methods to promote learner understanding, and the development of learning tasks. The content validity was found to be very appropriate at 1.0 for all questions except one (0.8) out of a total of six questions. The class strategy is the process of checking the learning objectives and tasks on the learning topic, the suitability of teaching and learning methods, whether students perform effective tasks, the effectiveness of learning objectives, and evaluating appropriate class management [43].

The teaching strategy should be constructed in an effective way to teach the values of the psychological, cognitive, and affective areas that students should learn in physical education classes [44]. Teachers should be able to teach the value of physical education through the physical education class model, teaching style, and teaching strategy [45]. Therefore, a number of questions were organized to confirm whether students achieved effective academic achievement in this study. Observation and interaction consist of checking students' task performance and degree of progress, providing appropriate feedback, determining whether to use various questions, inducing active and cooperative classes, inducing interaction, and providing fair educational opportunities to all learners. The content validity was found to be very appropriate for all seven questions, with a score of 1.0. Maintaining the learning environment consists of the appropriateness of the class site, space, and learning contents, the efficiency of the organization, efficient time manage-

ment, sufficient activity time, inappropriate behavioral guidance, and continuous safety inspection. The learning environment helps learners understand the learning content and effectively participate in this activity [40]. Many teachers and prospective teachers think that it will remain natural if a learning environment is created [15,17]. However, because of various variables, it is difficult to operate in the planned direction. Therefore, it is necessary to continuously check whether the learning environment is operating smoothly based on the various situations of the class.

According to previous studies, the development stage emphasizes the teacher's questions, feedback, opportunities for learning activities, validity between learning goals and activities, learners' individual differences, learning organizations, and the efficient use of time [2]. Silverman [46] suggested that the evaluation of interaction with learners and respect for opinions, learner behavior management, teaching strategy utilization, feedback, the induction of learning participation, efficient time acquisition, and equal and fair evaluation should be conducted in class. The reflective evaluation tool also maintains the same context as the teaching function suggested in previous studies. However, it is characterized by emphasizing the effects of the teacher's behavior and the learning environment on the safety of physical education classes and is composed of questions and evaluation criteria that reflect the real field. Physical education classes are run by various physical activities. Therefore, it is important to provide safety education to students. However, teachers are not properly conducting safety-based education [47,48]. Thus, this study included the evaluation question to strengthen awareness of the safety of physical education classes. The development stage smoothly connects the practice process from class preparation and evaluates the role of teachers.

Fourth, the "organizing stage" evaluates the competency to efficiently evaluate repetitive actions at the end of the class and the competency to guide reflective activities of the class. The organizing stage of the learning objective aims to check whether the learning goal has been achieved by looking back on the class with students [49]. However, there has been a strong perception that the organization stage of the class is a very simple process of conducting cool down exercises or previewing the next lesson. For this reason, there was a problem in that the necessary instruction was not properly conducted at the organization stage and was neglected. This stage consists of routine activities, summary, and closure. The content validity showed that all questions were very appropriate at 1.0. The routine activities consist of questions about organizing exercises, the organization of learning materials, and identifying patients. The summary and closure stage focused on evaluating students' understanding of learning goals, achievement, and the activity participation process to provide appropriate information. In this study, the conclusion stage emphasizes processes, such as the organization of materials, the achievement of learning content, feedback, student encouragement, and previewing the next lesson. These contents showed high validity in expert opinions and confirmed the educational meaning of the conclusion stage.

Fifth, the evaluation tool was developed for the purpose of identifying one's level of teaching competency, improving problems, and professional development through the results of the evaluation. Reflection in pre-service teacher education refers to the process in which pre-service teachers contemplate and judge the connectivity, relevance, and effectiveness of knowledge, performance, beliefs, and results [50]. Therefore, the reflective activities of pre-service teachers help to improve teaching methods and classes, teacher beliefs, confidence, and critical and creative thinking competencies [9]. In most pre-service teacher education programs, reflective activities are carried out through the instructor's feedback and reflection journal. However, unfortunately, rather than providing feedback through objective evidence based on certain criteria, it relies on the evaluation of selected evaluator criteria. Moreover, there is a limit in its ability to effectively evaluate the entire course of the class. However, this evaluation tool was evaluated based on systematic criteria from the class preparation stage to the organization stage, so the objectivity of evaluation could be secured. The evaluation criteria for each question enable reflective activities

through self- and peer evaluation. This study presented a new perspective that applies both qualitative and quantitative evaluation methods that include the objectivity of evaluation and the value of the reflective perspective. It is well-known that the various feedback and self-reflection of colleagues were very effective in enhancing the teaching skills of pre-service teachers [51]. Therefore, it is expected that the use of reflective evaluation tools, such as self- or peer evaluation developed in this study, can contribute to effectively enhancing the teaching competency of pre-service teachers. In summary, it can be seen that the core teaching skills that pre- or in-service physical education teachers must have should be the same. Therefore, in pre-service teacher education, it is necessary to focus on educating teachers on core teaching competencies that are practically used in the field. The problem with pre-service teacher education is that it does not properly reflect the characteristics required in the field, so it cannot demonstrate the practical teaching competencies used to guide students in the actual classroom [15]. As a result, they experience the shock of transition between practice and theory, leading to a fear of teaching a class, which becomes an obstacle to growing as an expert. For this reason, the use of the reflective evaluation tool developed in this study focuses on acquiring practical teaching skills that can be integrated with theory and practice. It can be a starting point to reduce the gap between theory and the field and increase the connection with teacher education programs. Evaluation tools are specifically and comprehensively produced and focus on teacher growth [52]. Therefore, this evaluation tool is expected to be of great help in enhancing the expertise and efficiency of pre-service sports teacher education programs.

Recently, Kwon [18] and Nadeem and Rahman [53] suggested that teacher competencies are not naturally developed in teaching and learning environments, but that continuous efforts are needed to specifically target and learn concepts, methods, and technologies for each teaching competency. However, more exploration is needed regarding the possibility of sustainably developing teaching competencies in the context of professional practice. The main importance of improving teaching competency is to focus on building the major, sustainable teaching competencies through teacher education programs [54]. Therefore, teacher education programs must focus on effective education for prospective teachers for their professional development [55,56]. In this study, the reflective evaluation tool used to evaluate the teaching competency of pre-service physical education teachers indicates that detailed educational experience in the pre-service teacher education stage should lead to practical competency. This evaluation tool is designed to systematically evaluate the key teaching skills required in the course of the class preparation–introduction–development–conclusion stages and the evaluation criteria for each question. With this tool, we attempted to increase the objectivity of evaluation by applying both quantitative and qualitative evaluation methods. The strength of this evaluation tool is that it specifically presents evaluation criteria for each question. Pre-service teachers are expected to be able to specifically cultivate sustainable teaching skills by planning and operating classes according to the evaluation criteria of this study. In particular, elementary, middle, and high school teachers, educational experts, and leaders are organized to enable self-evaluation or peer evaluation regardless of time or place, which will also help to revitalize reflective teaching activities.

The evaluation tool presents the basic teaching behaviors required in the process of physical education class. However, it is burdensome to evaluate 47 questions. Many questions lower the respondents' probability of sincere responses while placing a burden on the assessment. This evaluation tool was developed for the purpose of pre-service teachers' development. The purpose of the evaluation tool is to learn the basic teaching skills in the detail necessary for physical education teachers. This tool specifically presents various teaching behaviors necessary in physical education classes. In a study that evaluated the teaching performance of physical education teachers, Kim [15] confirmed that it took a lot of time to evaluate 50 questions, and that it was difficult to evaluate questions with low relevance to the class. Therefore, the items should be selectively utilized based on the

purpose of using the evaluation tool. It can also be beneficial to consider learning activities or class places.

The evaluation tool developed in this study can be said to be valuable as a tool to systematically accumulate practical teaching experiences that can reduce the gap between theory and practice in pre-service teacher education. However, since this evaluation tool was developed based on the situation with educational experts in Korea, it is necessary to consider the possibility that there may be differences in teaching competency evaluation according to the characteristics of society, culture, education system, and learning environment. Although the context of the core teaching function of teachers required in physical education classes is the same, the questions can be reconstructed and used according to the purpose and situation of pre-service teacher education in various countries. This study is meaningful in that it is a study that has developed a comprehensive evaluation tool that can cultivate teaching competency for prospective teachers who have neglected teacher evaluation research. Additionally, classes can be used as learning tools to specifically and systematically reflect. Therefore, it is necessary to actively introduce the possibility of the continuous use of the reflective evaluation tool developed in this study by elementary, middle, and high school teachers, as well as pre-service teacher education institutions and programs at universities. The adoption of a sustainable education professional development program that provides various benefits is very important and could contribute to effective teacher education.

## 6. Conclusions and Suggestions

The purpose of this study was to develop a reflective evaluation tool and use it in education as a method to improve the teaching competency of pre-service physical education teachers. A Delphi survey was conducted to modify the questions based on the existing physical education teacher evaluation tools and evaluation criteria were developed for each question. The evaluation tool consists of 46 questions in the four stages of class preparation (learning environment creation), introduction (routine activities, learning goals, and task presentation), development (class strategy, observation and interaction, and maintenance of the learning environment), and conclusion (routine activities, summary, and closure). In addition, the evaluation criteria for each question were developed to increase the accuracy of the evaluation, identify the strengths and weaknesses of the class through a reflective journal, and seek improvement measures. Finally, an evaluation tool including quantitative and qualitative methods to be used in the education of prospective physical education teachers was developed.

The educational conclusions of this study are as follows. First, a reflective teaching competency evaluation tool was developed to evaluate the core teaching behavior required of physical education teachers in the field. It was found that what pre-service and in-service teachers commonly regard as important is practical competency. Therefore, pre-service teacher education should be conducted based on practice so that the teaching competency required in the school can be cultivated. Second, a reflective evaluation tool was designed to present specific evaluation criteria for teaching behavior and to perform reflective activities. It will help to understand the correct standards of teaching behavior required by physical education teachers and to cultivate the competency to effectively perform the class plan–execution–reflection process.

Based on the results of this study, we make two suggestions for the use of reflective evaluation tools in pre-service teacher education programs. First, research should be conducted to explore educational meaning using reflective teaching competency evaluation tools in pre-service teacher education. In this study, an evaluation tool for pre-service teachers was developed. Therefore, research is needed to confirm the possibility of use in pre-service teacher education by using newly developed evaluation tools. Second, research on the development of various learning tools and educational programs should be conducted to enhance the teaching competency of pre-service teachers. There has not been much interest in the development of educational materials, such as educational programs

and learning tools used to cultivate teaching competency in pre-service teacher education. Most of the studies focus on in-service teachers. However, teaching competency requires thorough teacher training during pre-service teacher education. If one has the teaching competency before starting as a teacher, one will be able to provide good physical education classes to students while minimizing the difficulties as a first-time teacher.

**Author Contributions:** Conceptualization, C.-M.K.; methodology, C.-M.K. and E.-C.K.; writing— original draft preparation, C.-M.K.; writing—review and editing, E.-C.K.; visualization, C.-M.K. and E.-C.K.; supervision, E.-C.K. All authors have read and agreed to the published version of the manuscript.

**Funding:** This research was funded by the National Research Foundation of Korea (NRF-2020S1A5B5A17091060).

**Institutional Review Board Statement:** Not applicable.

**Informed Consent Statement:** Not applicable.

**Data Availability Statement:** Not applicable.

**Conflicts of Interest:** The funders had no role in the design of the study; in the collection, analyses, or interpretation of data; in the writing of the manuscript; or in the decision to publish the results.

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
