# Peer review of "An Exploration of a Reflective Evaluation Tool for the Teaching Competency of Pre-Service Physical Education Teachers in Korea"

_sustainability, doi:10.3390/su14138195_

Round 1
Reviewer 1 Report
The study presented is well organized and written. However, there are some issues/suggestions to resolve:
- Abstract: what is the gap that this study aims to solve? The methodology used is not clearly understood.
- The authors decided to present the introduction and literature review together. However, I am of the opinion that these two sections should be separated in order to first frame the topic and then review other studies. In the literature review, research questions should be formulated to clearly understand the purpose of this study. The literature review must still be in-depth and updated (few references and are very old).
- In the methodological part, point 2.2 is missing. or the sub-sections are wrongly numbered. These subsections must be supported by literature on the Delphi method. The reason for using this methodology is also not clear. It was expected that they would demonstrate that this methodology is the most appropriate to achieve the objective of this study.
- Discussion of results should be thoroughly reviewed. It lacks more foundation and articulation with the literature review.
Author Response
"Please see the attachment

Reviewer 2 Report
Overall, this is a good-written paper. Just a few comments for you.
1. The purpose of your study is to develop an evaluation tool that can be used in the gym contexts instead of the general classrooms (lines 104 & 142). While you describe the developmental processes of this tool in a detailed manner, you did not describe (1) if it is good to use it in the different gym contexts (e.g., swimming pool, track field, gym), (2) which gym contexts you would like to implement this tool? (3) or if this tool fits all physical education contexts? Please add more information about it in the discussion section and/or limitation section.
2. This survey tool was developed through the Delphi method and comments were provided by experts (professors and physical education teachers). It would be good to cover the comments about how pre-service physical education teachers perceive this survey tool as well. Please add more information about it in the limitation section as well.
3. Please update your references if possible. Also, check how you may use doi to track your reference.
Reviewer 3 Report
The article is innovative and provides data of great interest in the corresponding field.
Author Response
Thank you very much for your kind suggestion and recommendation. Thank you very much for your serious criticism and suggestion that our paper has improved much professionally and academically.
Round 2
Reviewer 1 Report
The authors made all suggestions. Thanks.